# Anti-HIV and Anti-Candidal Effects of Methanolic Extract from *Heteropterys brachiata*

**DOI:** 10.3390/ijerph18147270

**Published:** 2021-07-07

**Authors:** Maira Huerta-Reyes, Luis O. Sánchez-Vargas, Getsemaní S. Villanueva-Amador, Luis A. Gaitán-Cepeda

**Affiliations:** 1Unidad de Investigación Médica en Enfermedades Nefrológicas, Hospital de Especialidades “Dr. Bernardo Sepúlveda Gutiérrez”, Centro Médico Nacional Siglo XXI, Instituto Mexicano del Seguro Social, Ciudad de México 06720, Mexico; chilanguisima@yahoo.com; 2Laboratorio de Bioquímica y Microbiología Oral, Facultad de Estomatología, Universidad Autónoma de San Luis Potosí, San Luis Potosí 78290, Mexico; lo.sanchezvargas@gmail.com; 3Departamento de Medicina y Patología Oral y Maxilofacial, División de Estudios de Postgrado e Investigación, Facultad de Odontología, Universidad Nacional Autónoma de México, Circuito Institutos s/n, Ciudad Universitaria, Coyoacán, Ciudad de México 04510, Mexico; purpureakush@ciencias.unam.mx

**Keywords:** HIV, AIDS, candidiasis, antiretroviral treatment, antifungal treatment, *Heteropterys*, *Malpighiaceae*

## Abstract

Nowadays, the HIV pandemic is far from controlled. HIV+/AIDS patients show a serious risk of developing resistance to HIV antiretroviral drugs and to be orally colonized by *albicans* and non-*albicans Candida* strains resistant to antifungals. As a consequence, new drugs that possess anti-candidal and anti-HIV effects would represent an alternative in the comprehensive treatment of HIV+/AIDS patients. The present study evaluates the possible anti-HIV and anti-*Candida* effects of a methanolic extract from *Heteropterys brachiata* (Hb MeOH), an American tropical plant. The anti-HIV effect of Hb MeOH was tested using a non-radioactive colorimetric method (Lenti RT^®^ Activity Assay; Cavidi Tech) that uses reverse transcriptase of HIV-1 enzyme as enzymatic target. The anti-candidal effect of HbMeOH extract was evaluated by following a standardized test protocol of microdilution for yeast using the *Candida albicans* strain ATCC^®^ 90028. The Hb MeOH at 1 mg/mL concentration shows 38.5% RT-HIV inhibition, while Hb MeOH at 10 mg/mL concentration produced 98% *C. albicans* growth inhibition. Our findings show that the Hb MeOH possesses a strong anti-candidal activity and moderate anti-HIV effect and suggests that the plant extract could be considered as a potential candidate for HIV/AIDS treatment.

## 1. Introduction

In 2021, the pandemic caused by the human immunodeficiency virus (HIV) has been around for 40 years [1]. In this frame time, a great amount of information has been accumulated on HIV and the acquired immune deficiency syndrome (AIDS). Consequently, the knowledge about the etiopathogenesis and natural history of the infection, as well as its prevention and the number of available treatments, has increased. In spite of these efforts, the HIV pandemic is far from controlled and its eradication is not expected in the short-term. Currently, there are about 38 million HIV-positive people worldwide [2]. Despite advances in antiretroviral treatment (ART), and the emergence of new antiretroviral drugs, HIV presents several challenges that must be overcome. One of the most important is the duration one must be on ART, since, regardless of HIV type and the number of drugs administered, one must take ART for a lifetime, which increases the possibility of an infected person developing resistance to ART [3]. Another crucial aspect is that ART drugs are not equally available to all HIV+ patients. It is estimated that only 68% of all patients with HIV/AIDS have access to therapy or are on ART [2].

Oral candidiasis (OC) is the most frequent oral opportunistic infection in HIV+/AIDS people [4,5]. Approximately 80% of all HIV+/AIDS patients will present with at least one clinical event of OC in their lives and due to their immunodeficiency condition, they will present with continuous reinfections or reactivation of resident strains. Thus, these patients will need to undergo recurrent antimycotic treatments, which will increase the presence of *albicans* and non-*albicans Candida* strains resistant to antifungals in their bodies and will also increase the colonization of species with intrinsic resistance such as *C. glabrata*, *C. dubliniensis,* and *C. auris* [6,7,8].

The aforementioned issues speak to the importance of the development of effective anti-HIV and antifungal drugs with minimal side effects to prevent the spread of antimicrobial resistance. There have been attempts to identify drugs from plants with dual effects as a possible alternative. Different species of *Heteropterys* have shown pharmacological activity. For example, it has been reported that *H. tomentosa* possesses a double anti-viral effect against poliovirus type 1 and bovine herpes type 1 [9]. Phytometabolites, such as tannins [10,11], flavonoids [12], terpenes [13], polyphenols [14], and saponins [15], obtained from the family Malpighiaceae have been reported to have in vitro anti-candidal activities. The Malpighiaceae family is composed of approximately 75 genera and 1300 species distributed in the American continent. Mexico is considered the center of origin and diversification of this plant family [16]. In the genus *Heteropterys,* which belongs to the Malpighiaceae family, the presence of tannins and flavonoids have been previously identified in the methanolic extracts for treatment of mental diseases by our research group [17].

The main objective of this study is to discuss the anti-HIV and anti-*Candida* effects of a methanolic extract from *H. brachiata*. These new compounds have been found to offer alternative and possibly beneficial, cost-effective forms of antiretroviral therapy.

## 2. Materials and Methods

### 2.1. Obtaining the Methanolic Extract of Heteropterys brachiata

The *H. brachiata* extract was collected from Quilamula, Huatla, Morelos in Mexico on 4 October 2018. A backup of the extract was collected and sent to the National Herbarium of Mexico’s Institute of Biology, UNAM (MEXU). The extract was obtained through the utilization of standardized methods [18]. First, the aerial parts of the plant were dried at room temperature in a dark room for 15 days. Once the plant material was dry, it was ground with a mechanical mill (the weight of the dried and ground plant material was 2.200 kg) and dewaxed with *n*-hexane for 24 h. Subsequently, the extraction was carried out (3×) overnight by mixing it with 100% methanol. A volume of 7.5 L of liquid with methanol was extracted per kilogram of plant material. The plant material was filtered and the liquid part was dried at a high vacuum level with a rotavapor (Rotavapor^®^ BUCHI R-100), leading to the production of methanolic extract from *Heteropterys brachiata* (Hb MeOH). The chemical composition of Hb MeOH was characterized and reported by our research group in a previous manuscript [17] and in the patent [18]. The main constituents of the Hb MeOH are chlorogenic acid (3.2 mg/kg) and chlorogenic acid methyl ester (60 mg/kg).

### 2.2. Evaluation of the Anti-HIV Effect of Methanolic Extract from Heteropterys brachiata

The anti-HIV effect of Hb MeOH extract was assessed using a non-radioactive colorimetric Lenti RT^®^ Activity Assay (Cavidi Tech, Uppsala, Sweden) that uses HIV-1 reverse transcriptase (RT) enzyme as the enzymatic target. We followed the instructions of the manufacturer. Hb MeOH extract of different concentrations was added to each well of the 96-well microtiter plate along with a reaction mixture, containing primer, a nucleotide (5-bromo-3-deoxyribouridine 5′-triphosphate), and buffers. Later, the RT enzyme was added. If there was no inhibition of enzymatic activity, the RT synthesized the DNA. Later, the anti bromo-deoxyribouridine (α-BrdU) antibody conjugated to alkaline phosphatase was placed in all the wells of the microtiter plate. The result was reported as percentage of RT inhibition. The Hb MeOH extract was probed at concentrations of 1 mg/mL, 0.1 mg/mL, and 0.01 mg/mL in an anti RT VIH kit. To obtain these concentrations, Hb MeOH extract was dissolved in DMSO. The experiments were performed in triplicate. Nevirapine (Viramune^®^), a transcriptase reverse inhibitor, was used as a positive control.

### 2.3. Evaluation of the Anti-Candidal Effect of Methanolic Extract from Heteropterys brachiata

To establish the anti-candidal effect of Hb MeOH, a standardized test protocol [19] using the *Candida albicans* strain ATCC^®^ 90,028 was performed. The stock solution from the Hb MeOH extract was 1 g/mL after using DMSO as a dissolvent. Later, from the stock solution, dilutions were made to obtain concentrations 100 times higher than the final concentrate. A second dilution (1/50) was then performed by taking 100 µL of each concentration and adding it to 4.9 mL of Roswell Park Memorial Institute (RPMI) 1640 (Gibco^®^) medium.

The *C. albicans* inoculum was prepared in the following way; *C albicans* strain was incubated at 25 °C for 24 h on Sabouraud dextrose/chloramphenicol agar plates (CRITERION ™, HardyDiagnostics, Santa Maria, CA, USA). Later, Five colonies ≥ 1 mm were taken and incubated with yeast peptone dextrose broth at 37 °C for 48 h. After incubation, a cell pellet was obtained by centrifugation (34 g/10 min). The nutrient broth was removed and then 5 mL of 0.01 M phosphate-buffered salt (PBS) was added. The presence of fungal cells was verified by observation under an optical microscope (Leica DM500, objective 40×). An aliquot of 30 µL was taken by adjusting the inoculum to an optical density of 1–5 × 10^3^ CFU/mL and placed in a tube with 700 µL of PBS (2.7 mM potassium chloride and 137 mM sodium chloride, pH 7.4; Sigma-Aldrich, St. Louis, MO-IL, USA) to obtain a total volume of 1 mL. Then, 630 µL of inoculum was added to 62,370 mL of RPMI. The inoculum was stored until it was ready to be used. The microtiter plates were filled with 100 µL of inoculum and 100 µL of the solution to be tested. The final concentrations of the extract were: 10 mg/mL, 5 mg/mL, 2.5 mg/mL, 1.25 mg/mL, 625 µg/mL, 312 µg/mL, 157 µg/mL, 78 µg/mL, 39 µg/mL, and 19 µg/mL. Columns of the microtiter plates (numbers 1 and 12) served as negative and positive controls, respectively. Number 1 contained 200 µL of RPMI medium and number 12 contained 100 µL of RPMI with 100 µL of inoculum. Plates were incubated at 37 °C for 24 h. Fluconazole (Flucoxan^®^) was used as the reference antifungal to validate the standardization of the M27-A3 method, and for the interpretation of cut-off points. Furthermore, a comparison of the sensitivity or resistance of the *C. albicans* inoculum was made between the Hb MeOH and chlorogenic acid (Sigma-Aldrich, CAS 327-97-9) using standard dilutions.

In both cases, reading of the plates was performed in a microtiter plate reader with a length of 405 nm using the SKANTI software 3.1, the research editor for Multiskan FC (Thermo Scientific, Waltham, MA, USA). Inhibition is reported as the percentage of inhibition. The research protocol was authorized by an ethical committee of participant institutions.

## 3. Results

### 3.1. Anti-HIV Effect of Hb MeOH Extract

The methanolic extract from *H. brachiata* at a concentration of 10 mg/mL showed a 38.5% inhibition, while a concentration of 1 mg/mL showed a 25.8% inhibition. At a concentration 100 µg/mL, the percentage inhibition was 12.5%.

### 3.2. Anti-Candidal Effect of Hb MeOH Extract

Following the Clinical & Laboratory Standards Institute (CLSI) guidelines that state that a strain is considered sensitive to a substance tested if the percentage if inhibition is greater than 50%, the *C. albicans* strains tested showed sensitivity to the methanolic extract from *H. brachiata*.

At a concentration of 10 mg/mL, the Hb MeOH extract produced 98% inhibition, the highest percentage of inhibition obtained. At a concentration of 5 mg/mL, we observed an inhibition of 62%. The lowest concentration at which inhibition was obtained (61%) was 2.5 mg/mL. Table 1 shows the inhibition percentages obtained at different concentrations of Hb MeOH and chlorogenic acid. The highest inhibition percentage obtained using chlorogenic acid was 15%.

## 4. Discussion

The study evaluated the anti-candidal and anti-HIV-1 effects of a methanolic extract from *Heteropterys brachiata* as a possible alternative treatment for HIV+/AIDS patients. The medicinal plant extracts are safe, cheap, and effective [20]. Our findings showed that the methanolic extract from *H. brachiata* has moderate anti-HIV inhibitory properties. To the best of our knowledge, this is the first time that this extract has been reported as a potential treatment for HIV. We observed a percentage inhibition of RT HIV-1 considered moderate-acting, similar to that reported in other plant extracts [21]. Phytochemical metabolites, such as alkaloids, flavonoids, phenolic compounds, glycosides, tannins, and saponins, protect against the main enzymatic mechanisms of HIV, for a review see [22,23], and particularly, alkaloids [24], saponins [25], and tannins [26] have already been identified as inhibitors of HIV-1 TR. Due the RT HIV-1 moderate-acting inhibition reported in our work, future anti-HIV investigations may consider escalating the metabolites detected in the HbMeOH extract to see if they can obtain a higher percentage of inhibition, and a variety of HIV targets for HbMeOH testing.

Chlorogenic acid was identified as one of the main components in the Hb MeOH extract in a previous report from our research group [17]. Compounds derived from chlorogenic acid have been reported to be potent inhibitors of HIV integrase by a number of studies [27,28]. In the case of the enzyme HIV-TR, recent research showed that eight chlorogenic acid derivatives exhibited HIV-1 TR inhibition [29]. These data allow us to suggest that compounds derived from chlorogenic acid may represent a source of moderate HIV-1 RT inhibitors. Other authors have focused on the structural modification of chlorogenic acid in order to potentiate its anti-HIV properties through the inhibition of the α-glucosidase. Besides the role of the inhibitors of α-glucosidase in the progression of diabetes by decreasing carbohydrate digestion and absorption, in recent years, the inhibition of α-glucosidase has been considered a promising strategy for the development of novel anti-HIV agents due to the glycosylation of viral envelope glycoproteins [30]. The envelope glycoproteins gp120 and gp41 play crucial roles in HIV entry, thus serving as key targets for the development of HIV entry inhibitors [31]. Inhibitors of α-glucosidases can cause glycoproteins to misfolded and remain within the endoplasmic reticulum, interfering with the viral life cycle and infectious process [32]. Thus, Hattori et al. [30] proposed the utilization of novel α-glucosidase inhibitors through the optimization of the hydrophilicity of chlorogenic acid by addition of alkyl chains of various lengths. The addition of ketal or acetal bonds to chlorogenic acid derivatives caused an increase in the inhibitory activity of 1 α-glucosidases. Chlorogenic acid and its derivatives, constitute novel anti-HIV agents.

In HIV+/AIDS patients, including those on antiretroviral therapy, the most common oral opportunistic infection is Oropharyngeal candidiasis [4,5]. The presence of OC is an important prognostic value in HIV/AIDS patients [33], therefore medical control of oral candidiasis is essential. However, there are several obstacles that have prevented the achievement of this goal. On one hand, there is a limited amount of suitable, effective, and universally affordable care for people living with HIV/AIDS in developing countries [34]. On other hand, there is an increase in the prevalence of drug-resistant *Candida*, and the emergence of other inherently drug-resistant *Candida* species [7]. Therefore, the development of an alternative, efficient antifungal drug is necessary and essential [35]. Several plants possess effective anti-candidal properties. *Cinnamomum zeylanicum*, eucalyptus, lemongrass oil, ginger grass oil, peppermint, coriander, and *Thymus villosus* all possess anticandidal properties [36]. Our findings showed that the Hb MeOH at a concentration of 2.5 mg/mL, reaching a maximum inhibition (98%) at a concentration of 10 mg/mL possesses properties against *C. albicans* (61% inhibition). Interestingly, the compound chlorogenic acid, which has been described before as one of the main compounds in Hb MeOH [17], showed a 15% inhibition when isolated, representing ≤25% of inhibition when compared with the crude extract. Consequently, our results reveal that the crude plant extract is a powerful medicine [37]. In addition, anti-candidal metabolites derived from the Malpighiaceae family such as tannins [10,11], flavonoids [32], terpenes [13], polyphenols [14], and saponins [15], have been reported. The phytochemical screening of *Heteropterys brachiata* extract exhibited the presence of alkaloids, saponins, and tannins. Nevertheless, the identification of the possible and specific metabolites related to the antiviral and anticandidal effects is beyond the objective of this study, thus further research to elucidate this effect is necessary.

For HIV+/AIDS patients, the existence of a potential alternative drug with dual therapeutic effect is of utmost importance. Our results support this possible alternative since the methanolic extract from *H. brachiata* showed strong anti-candidal activity and moderate anti-HIV activity. There are two precedents in this regard; both emerged since the introduction of protease inhibitors of HIV-1 (PI-HIV-1) to antiretroviral therapy and the consequent establishment of combined antiretroviral therapy (cART) [38]. cART decreases the viral load of HIV-1 and has a direct effect on the treatment and control of other opportunistic infectious diseases [39].

A decrease in oral candidiasis has been reported, as well as a regression of Kaposi’s sarcoma related to HIV (KS-HIV). In the case of OC, cART produced a dramatic decrease, especially in the pseudomembranous clinical form. The decrease in OC prevalence was attributed to improvements in immune response (increase of CD4+ lymphocytes cell count). It was reported that strains of *Candida* spp. isolated from HIV+ people, were susceptible to saquinavir (a protease inhibitor) and indinavir, with 97% and 52% susceptibility, respectively [40]. Regarding Kaposi sarcoma, it has been reported that PI-based ART or NNRTI-based ART significantly decreased incidence rates by at least 10-fold among HIV+/AIDS people who received no ARVs compared to those receiving ART [41]. Even though the effects of cART on HHV-8 have not been fully elucidated, PI appears to possess a direct antiangiogenic property [42].

The possibility of the existence of a potential alternative anti-HIV drug with an anti-candidal effect was discussed in this paper. Although these are empirical results that need to be confirmed experimentally, they show the possibility that an extract from plants with a double therapeutic effect can be used to treat both HIV and OC. For a disease like HIV/AIDS where there is an enormous demand for ART drugs, advances in phytopharmacy could help close the gap. Technological development and the standardization of procedures for the production and purification of natural extracts, will make this possible.

## 5. Conclusions

Our findings show that methanolic extract from *H. brachiata* is a potential alternative to ART for HIV treatment and treatment for OC. Given the low cost of production, it is cost effective and affordable. The crude extract possesses a four-times-higher inhibition of *C. albicans* than pure compound chlorogenic acid.

## Figures and Tables

**Table 1 ijerph-18-07270-t001:** Inhibition of *Candida albicans* observed at different concentrations of methanolic extract from *Heteropterys brachiata* and chlorogenic acid.

	*Hb* MeOH (mg/mL)					CHLOROGENIC ACID (µg/mL)				
	10	5	2.5	1.25	0.625	17.5	8.75	4.37	2.18	1.09
Percentage of inhibition	98%	62%	61%	15%	22%	15%	0%	12%	4%	0%

*Hb* MeOH = methanolic extract of *H. brachiata*.

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
