# Peer review of "Anti-HIV and Anti-Candidal Effects of Methanolic Extract from Heteropterys brachiata"

_ijerph, 2021, doi:10.3390/ijerph18147270_

Round 1

Reviewer 1 Report

The manuscript by Huerta-Reyes et al. focuses on the potential of a plant alcoholic extract as anti-HIV and anti-Candida agent, the latter being stronger than the anti-viral effect of the extract. Although the topic is of great interest for our health, the authors failed to provide crucial information regarding the Hb MeOH extract here used, such as its physical-chemical characterization, to corroborate the anti-viral and anti-fungal activity observed. Moreover, although this extract appears to be highly active against Candida spp. strains, its efficacy drastically decreased (38.5%) as anti-HIV agent, making me to wonder if this can be actually considered a good anti-HIV choice. 

Thus, I cannot recomend the present manuscript for publication in its present form. I could reconsider it if the authors characterize their product.

Below my comments:

GENERAL COMMENTS:

Introduction:

Page 1, lines 61-65: I think the authors should expand on this subject, which actually describes the main purpose of their work. Most of the introduction regards HIV and AIDS, yet the authors should educate the reader regarding why choosing plants, why those of Heteropterys, etc… For instance, the sentence reported on page 1, lines 72-74 should be moved from section 2.1 to the above section of the introduction.

A crucial section missing from this paper is that regarding the characterization of the Hb MeOH from a physical-chemical point of view. I strongly suggest the authors to perform, at least, FTIR spectroscopy on this product, although they should also think about evaluating the actual chemical composition of the extract by means, for instance, of HPLC and mass spectrometry to identify potential candidates exerting anti-viral and anti-fungal activity. Simply stating that a compound, or a mixture of diverse substances, has anti-viral and anti-candidal effect without explaining why is not enough; at least the authors should provide some information regarding the chemical nature of the plant extract used.

Discussion section needs to be strongly implemented from both a biological and chemical point of view.

SPECIFIC COMMENTS:

Page 1, line 24: Please, use italics when referring to Candida albicans.

Page 1, line 25: Please, remove “Results:”; moreover, the authors should express the concentration of HbMeOH either as percentage or as mg/ml to improve the clarity of the data.

Page 1, line 34: Please, change “In the current year 2021” with “In 2021”.

Page 1, line 37-39: Please, reformulate this sentence to improve its clarity.

Page 1, line 43: Please, change “One of the most important regards to its duration…” with “One of the most important is represented by its duration…”

Page 1, line 52: Please, substitute “and they…” with “they…”.

Page 1, lines 53-56: Please, change this sentence with “Thus, these patients must undergo recurrent antimycotic treatments, which increase the presence…”.

Page 1, lines 57-59: Please, reformulate this sentence to improve its clarity.

Page 1, line 60: I think the definition of “transcendent” is different from what the authors are trying to say.

Page 2, lines 88-95 and 98-100: The explanation of the Lenti RT Activity Assay is too much general; for instance, when the authors write “an antibody conjugated to alkaline phosphatase is placed”, to which antibody are they referring to? Also, no amounts of reagents are here reported. Please, be more specific.

Page 2, line 97: Are the percentage of Hb MeOH extract used reported as % w/v? If that is the case, please include this information.

Page 3, lines 109-110: What do the authors mean with fluconazole was used as a reference? Reference of what…?

Page 3, line 115: Rpm are variable from one rotor to another, as they depend on the length arm. Please, provide the correspondent value converted using “g” as unit.

Page 3, line 116: How did the authors prepare PBS? Details are missing.

Page 3, lines 125-126: As before, what do the authors mean with “a comparison of sensibility Hb MeOH vs chlorogenic acid”?

Page 3, lines 134-137: H. brachiata should be in italics. Moreover, if there is a table listing the percentage of inhibition, these sentences are redundant. Either remove them or the table.

Page 4, lines 143-147: See comment right above.

Page 4, line 147: What does “NCCLS” stand for? Please, spell it out the first time it appears in the text.

Page 4, lines 149-150 and 153: C. albicans and H. brachiata should be in italics.

The authors should use the standard nomenclature of units; for instance, grams is abbreviated as g, not gr.

Author Response

“The manuscript by Huerta-Reyes et al. focuses on the potential of a plant alcoholic extract as anti-HIV and anti-Candida agent, the latter being stronger than the anti-viral effect of the extract. Although the topic is of great interest for our health, the authors failed to provide crucial information regarding the Hb MeOH extract here used, such as its physical-chemical characterization, to corroborate the anti-viral and anti-fungal activity observed. Moreover, although this extract appears to be highly active against Candida spp. strains, its efficacy drastically decreased (38.5%) as anti-HIV agent, making me to wonder if this can be actually considered a good anti-HIV choice”.

“Thus, I cannot recomend the present manuscript for publication in its present form. I could reconsider it if the authors characterize their product”.

Below my comments:

GENERAL COMMENTS:

Introduction:

“Page 1, lines 61-65: I think the authors should expand on this subject, which actually describes the main purpose of their work. Most of the introduction regards HIV and AIDS, yet the authors should educate the reader regarding why choosing plants, why those of Heteropterys, etc… For instance, the sentence reported on page 1, lines 72-74 should be moved from section 2.1 to the above section of the introduction.”

Answer: The paragraph, page 2, line 66 - 72, was rewritten highlighting because the genus Heteropterys is an attractive alternative to explore.

“A crucial section missing from this paper is that regarding the characterization of the Hb MeOH from a physical-chemical point of view. I strongly suggest the authors to perform, at least, FTIR spectroscopy on this product, although they should also think about evaluating the actual chemical composition of the extract by means, for instance, of HPLC and mass spectrometry to identify potential candidates exerting anti-viral and anti-fungal activity. Simply stating that a compound, or a mixture of diverse substances, has anti-viral and anti-candidal effect without explaining why is not enough; at least the authors should provide some information regarding the chemical nature of the plant extract used.”

Answer: Attending the instructions of the reviewer, we added the information in Materials and Methods section, about the chemical characterization of the methanolic extract of Heteropterys brachiata (Hb MeOH), describing the main components detected. This characterization was carried out by HPLC and our research team published before (Huerta-Reyes, M.; Herrera-Ruiz, M.; González-Cortazar, M.; Zamilpa, A.; León, E.; Reyes-Chilpa, R.; Aguilar-Rojas, A.; Tortoriello, J. Neuropharmacological in vivo effects and phytochemical profile of the extract from the aerial parts of Heteropterys brachiata (L.) DC. (Malpighiaceae). J Ethnopharmacol. 2013b, 146: 311-7.). The method for standarizing the obtention of this extract and its chemicals componentes were protected by patent (Huerta-Reyes M, Herrera-Ruiz M, Zamilpa-Álvarez A, González-Cortazar M, Tortoriello-García J, Aguilar-Rojas A, inventors; Extracto de Heteropterys brachiata, método de obtención y uso para el tratamiento de ansiedad y depresión. México. Patent 289104. June 29, 2011). Thus, since this information was already published and is extensive, we describe briefly the most important aspects in Materials and Methods section, and cited the original article and patent for a detailed technical description. Page 2, line 83-85, and 92 – 96.

“Discussion section needs to be strongly implemented from both a biological and chemical point of view.”

Answer: Following the comments of the reviewer, the Discussion section was improved in biological and chemical aspects. Page 5, lines 205 - 219. References 30,31 and 32 were added. 

“SPECIFIC COMMENTS”:

“Page 1, line 24: Please, use italics when referring to Candida albicans”.

Answer: Candida albicans was written in italics. Even more, the entire manuscript was meticulously reviewed in this respect.

“Page 1, line 25: Please, remove “Results:”; moreover, the authors should express the concentration of HbMeOH either as percentage or as mg/ml to improve the clarity of the data”.

Answer: “Results” was removed. The concentration of HB MeOH was homogenized in the entire manuscript.

“Page 1, line 34: Please, change “In the current year 2021” with “In 2021”.

Answer: It was delete “… the current year…” Line 35

Page 1, line 37-39: Please, reformulate this sentence to improve its clarity”.

Answer: The sentence was reformulated to improve its clarity.  Page 1, line 38 - 41.

“Page 1, line 43: Please, change “One of the most important regards to its duration…” with “One of the most important is represented by its duration…”

Answer: The sentence was change to “One of the most important is represented by its duration…” Page 1, line 44.

“Page 1, line 52: Please, substitute “and they…” with “they…”.

Answer: It was delete “and..” Page 2, line 53.

“Page 1, lines 53-56: Please, change this sentence with “Thus, these patients must undergo recurrent antimycotic treatments, which increase the presence…”.

Answer: The sentence was change with “Thus, these patients must undergo recurrent antimycotic treatments, which increase the presence…” Page 2, line 54 – 55.

“Page 1, lines 57-59: Please, reformulate this sentence to improve its clarity”.

Answer: The sentence was reformulated to improve its clarity. 

“Page 1, line 60: I think the definition of “transcendent” is different from what the authors are trying to say”.

Answer: The word “transcendent” was change for important. Page 2, line 61.  

“Page 2, lines 88-95 and 98-100: The explanation of the Lenti RT Activity Assay is too much general; for instance, when the authors write “an antibody conjugated to alkaline phosphatase is placed”, to which antibody are they referring to? Also, no amounts of reagents are here reported. Please, be more specific”.

Answer: As we originally wrote in the “Materials and Methods” section, “Evaluation of the anti-HIV effect of methanolic extract of Heteropterys brachiata” subsection, the anti-HIV effect of HbMeOH extract was evaluated by a commercial kit denominated Lenti RT® Activity Assay from the company Cavidi Tech. As regularly, methodology reports of kits only refer that authors follow the instructions of the manufacturer and offer a brief general description about it, which is our case. This style of the report has been widely employed and accepted in international literature since the detailed instructions are available on the kit or manufacturer web and are open to the public usually. As an example of this style of report, we cite next only 5 manuscripts from the numerous that could be found in the literature. Some of them also reported the same kit that we employed in the present manuscript.

- Chan, C. N., McMonagle, E. L., Hosie, M. J., & Willett, B. J. (2013). Prostratin exhibits both replication enhancing and inhibiting effects on FIV infection of feline CD4+ T-cells. Virus research, 171(1), 121-128.

In this article, please see 2.5 subsection. Authors are reporting the employ of the same kit than us.

- Bitzegeio, J., Sampias, M., Bieniasz, P. D., & Hatziioannou, T. (2013). Adaptation to the interferon-induced antiviral state by human and simian immunodeficiency viruses. Journal of virology, 87(6), 3549-3560.

In this article, please see “Virus replication assay” subsection. Authors are reporting the employ of the same kit than us.

Crespo, H., Bertolotti, L., Juganaru, M., Glaria, I., de Andrés, D., Amorena, B., ... & Reina, R. (2013). Small ruminant macrophage polarization may play a pivotal role on lentiviral infection. Veterinary research, 44(1), 1-13.

In this article, please see “Viral infections and RT activity determination” subsection.

- Perdomo, M. F., Hosia, W., Jejcic, A., Corthals, G. L., & Vahlne, A. (2012). Human serum protein enhances HIV-1 replication and up-regulates the transcription factor AP-1. Proceedings of the National Academy of Sciences, 109(43), 17639-17644.

In this article, please see “Cell-Based Assays subsection”.

- Dietrich, I., McMonagle, E. L., Petit, S. J., Vijayakrishnan, S., Logan, N., Chan, C. N., ... & Willett, B. J. (2011). Feline tetherin efficiently restricts release of feline immunodeficiency virus but not spreading of infection. Journal of virology, 85(12), 5840-5852.

In this article, please see “Virus replication assays”.

However, despite, we included some detail of Lenti RT Activity Assay (Cavidi ®). Page 3, line 105 – 106, and 108 – 110.

“Page 2, line 97: Are the percentage of Hb MeOH extract used reported as % w/v? If that is the case, please include this information”.

Answer: Hb MeOH is reported as w/v, and the paragraph was reformulated as follows: “The stock solution of the Hb MeOH extract was 1 gr/mL using DMSO as dissolvent. Later, from the stock solution, dilutions were made to obtain concentrations 100 times higher than the final. A second dilution (1/50) was then performed by taking 100 µL of each concentration and adding it to 4.9 mL of Roswell Park Memorial Institute (RPMI) 1640 (Gibco®) medium. Page 3, line 117 – 122.

The final concentrations of Hb MeOH extract were detailed in lines 135 – 137.

“Page 3, lines 109-110: What do the authors mean with fluconazole was used as a reference? Reference of what…?”

Answer: Because fluconazole is a well-established antifungal, it is frequently used as a control to compare the sensitivity and resistance exhibited by candida strains to other likely antifungal substances, therefore, Fluconazole was used as a reference antifungal to validate the standardization of the M27-A3 method. The sentence “Fluconazole (Flucoxan®) was used as a reference antifungal to validate the standardization of the M27-A3 method and for the interpretation of the cut-off points.

“Page 3, line 115: Rpm are variable from one rotor to another, as they depend on the length arm. Please, provide the correspondent value converted using “g” as unit”.

Answer:  The RPM unit was converted to “g” unit. Page 3, line 127.

“Page 3, line 116: How did the authors prepare PBS? Details are missing”.

Answer: Details about PBS preparation were incorporated. The sentence “(2.7 mM potassium chloride and 137 mM sodium chloride, pH 7.4; Sigma-Aldrich, St. Louis, USA)” was added.  Page 3, line 131 -132. 

“Page 3, lines 125-126: As before, what do the authors mean with “a comparison of sensibility Hb MeOH vs chlorogenic acid”?

Answer: The sentence was reformulated as follows: “Furthermore, a comparison of the sensitivity or resistance shown by the C. Albicans inoculum was made between Hb MeOH and chlorogenic acid (Sigma-Aldrich) using standard dilutions”. Page 3, line 142 – 144.

“Page 3, lines 134-137: H. brachiata should be in italics. Moreover, if there is a table listing the percentage of inhibition, these sentences are redundant. Either remove them or the table”.

Answer: The table was removed.

“Page 4, lines 143-147: See comment right above”.

Answer: In our opinion table 2 offers data no show in the text. We respectfully request the inclusion of the table. If it is possible, the table will be renamed table 1. 

“Page 4, line 147: What does “NCCLS” stand for? Please, spell it out the first time it appears in the text”.

Answer: The sentence “Clinical & Laboratory Standards Institute (CLSI) was included. Page 4, line 161.

“Page 4, lines 149-150 and 153: C. albicans and H. brachiata should be in italics”.

Answer: H. brachiata and Candida albicans were written in italics.  Page 4, lines 163, 164, 168.

“The authors should use the standard nomenclature of units; for instance, grams is abbreviated as g, not gr”.

Answer: The nomenclature of units was meticulously reviewed in the entire manuscript

Reviewer 2 Report

The manuscript evaluates the methanol extract of Heteropterys brachiata as a dual drug against HIV and oral candidiasis (OC). Evaluation of the action was performed in vitro.

The research topic is relevant. The resulting activity allows the extract to be considered as a possible remedy for the treatment of AIDS and associated OC. The origins of research on the extract of this plant lie in traditional medicine. The authors view the alleged drug as cheap and non-toxic, available to all segments of the population.

The work was done in good faith, the methods correspond to the task at hand. The introduction and discussion are relevant and adequate.

However, the presentation of the results has drawbacks that require clarification.

  1. Please specify which chlorogenic acid you used. This name refers to the various esters.
  2. The authors cite a link to their previous work on the determination of chlorogenic acid in plant material ("Chlorogenic acid was identified as one of the main components in Hb 176 MeOH extract according to previous reports from our research group [16]"). I would like to understand what is its content in the methanol extract of H. brachiata. If the presentation of such data is possible, then an explanation should be given to table 2 and its comments in the text. In this form, it is not clear how one can say that the action of the extract is not limited to the action of chlorogenic acid, but is a consequence of the synergism of the complex of natural substances in the extract. In Table 2, the extract is given in mg/ml and chlorogenic acid in μg/ml. Without clarifying this issue, it is impossible to speak about the greater efficiency of the extract in comparison with chlorogenic acid. The result in the Conclusion section is unfounded. ("The crude extract possesses four times higher inhibition on C. albicans than the pure compound chlorogenic acid, one of its major components.")
  3. Also explain why in Table 2, in two cases, the percentage of growth inhibition is negative (-11%). Is it about stimulating growth?
  4. Less significant note: Line 103: please correct "1 gr/mL" to "1 g/ml".

Author Response

Review 2.

“The manuscript evaluates the methanol extract of Heteropterys brachiata as a dual drug against HIV and oral candidiasis (OC). Evaluation of the action was performed in vitro.

The research topic is relevant. The resulting activity allows the extract to be considered as a possible remedy for the treatment of AIDS and associated OC. The origins of research on the extract of this plant lie in traditional medicine. The authors view the alleged drug as cheap and non-toxic, available to all segments of the population.

The work was done in good faith, the methods correspond to the task at hand. The introduction and discussion are relevant and adequate.

However, the presentation of the results has drawbacks that require clarification.

Please specify which chlorogenic acid you used. This name refers to the various esters.

The authors cite a link to their previous work on the determination of chlorogenic acid in plant material ("Chlorogenic acid was identified as one of the main components in Hb 176 MeOH extract according to previous reports from our research group [16]"). I would like to understand what is its content in the methanol extract of H. brachiata. If the presentation of such data is possible, then an explanation should be given to table 2 and its comments in the text. In this form, it is not clear how one can say that the action of the extract is not limited to the action of chlorogenic acid, but is a consequence of the synergism of the complex of natural substances in the extract. In Table 2, the extract is given in mg/ml and chlorogenic acid in μg/ml. Without clarifying this issue, it is impossible to speak about the greater efficiency of the extract in comparison with chlorogenic acid. The result in the Conclusion section is unfounded. ("The crude extract possesses four times higher inhibition on C. albicans than the pure compound chlorogenic acid, one of its major components.")

Also explain why in Table 2, in two cases, the percentage of growth inhibition is negative (-11%). Is it about stimulating growth?”

“Less significant note: Line 103: please correct "1 gr/mL" to "1 g/ml".

Thank you very much for your comments.

Attending the instructions of the reviewer, we added the information in the Materials and Methods section, about the chemical characterization of the methanolic extract of Heteropterys brachiata (Hb MeOH), describing the main components detected. This characterization was carried out by HPLC and our research team published before (Huerta-Reyes, M.; Herrera-Ruiz, M.; González-Cortazar, M.; Zamilpa, A.; León, E.; Reyes-Chilpa, R.;

Aguilar-Rojas, A.; Tortoriello, J. Neuropharmacological in vivo effects and phytochemical profile of the extract from the aerial parts of Heteropterys brachiata (L.) DC. (Malpighiaceae). J Ethnopharmacol. 2013b, 146: 311-7.). The method for standarizing the obtention of this extract and its chemicals componentes were protected by patent (Huerta-Reyes M, Herrera-Ruiz M, Zamilpa-Álvarez A, González-Cortazar M, Tortoriello-García J, Aguilar-Rojas A, inventors; Extracto de Heteropterys brachiata, método de obtención y uso para el tratamiento de ansiedad y depresión. México. Patent 289104. June 29, 2011). Thus, since this information was already published and is extensive, we describe briefly the most important aspects in the Materials and Methods section and cited the original article and patent for a detailed technical description.

With this information about the main components of the Hb MeOH, our comment in the text about that the chlorogenic acid is one of the main components of the Hb MeOH and our interest in evaluating isolated as we originally showed in Table 2 (now table 1), were based on the findings and chemical characterization published and patented before the present manuscript. These findings published were incorporated into the present manuscript as we mentioned before.

In order to clarify the possible role of chlorogenic acid in the activities observed in the present manuscript, we added more information in the Discussion section and re-wrote the Conclusion section.

Concerning the chlorogenic acid, in this work, we used chlorogenic acid (Sigma-Aldrich) as we specified in the “Materials and Methods” section, “Evaluation of the anti-candidal effect of methanolic extract of Heteropterys brachiata” subsection. This pure chlorogenic acid according to the manufacturer has the empirical formula C16H18O9, the CAS-number 327-97-9, and the formula weight 354.31 g/mol. With this information, it is possible to distinguish it from the rest of the compounds that can be considered in the chlorogenic acids complex or derivatives.

Less significant note: Line 103: please correct "1 gr/mL" to "1 g/ml".

Answer: This mistake was corrected.

We hope that the additional changes to the manuscript sufficiently address the points raised by the reviewer.

Round 2

Reviewer 1 Report

Although the authors explained the lack of characterization of the plant extract and the need for further studies to explain the higher antifungal activity of this extract as compared to other compounds, I still think that this manuscript should be improved before being accepted for publication. Given the additional information provided by the authors, it seems there is no need to characterize this extract, although I still think that techniques such as FT-IR and NMR would be valuable.

In my opinion, simply reporting the activity of bioactive molecules against viruses and fungi does not qualify a study for publication. If the authors already characterized the extract, its activity should be expected, considering the biomolecules within. I suggest the authors perform at least preliminary experiments (such as ROS determination, evaluation of membrane potential) to better justify this observation. Thus, I cannot consider this paper for publication in the present form.

Author Response

Thank you very much for your comments.

As we referred before, the chemical characterization of the methanolic extract of Heteropterys brachiata was carried out by some diverse chromatographic and spectroscopic techniques that you will find described in detail in the manuscript published and in the patent (Huerta-Reyes, M.; Herrera-Ruiz, M.; González-Cortazar, M.; Zamilpa, A.; León, E.; Reyes-Chilpa, R.; Aguilar-Rojas, A.; Tortoriello, J. Neuropharmacological in vivo effects and phytochemical profile of the extract from the aerial parts of Heteropterys brachiata (L.) DC. (Malpighiaceae). J Ethnopharmacol. 2013b, 146: 311-7). In these last documents, we not only analyzed chemically the main components of the extract but also, quantified them and established a rigorous method of extraction which was reproducible and filled the rigorous requirements of a patent. Thus, in effect, additionally to these techniques that we employed for the methanolic extract of Heteropterys brachiata, exists other than can be also useful for this same purpose; however, the international literature in the phytochemistry area are full of reports about techniques for characterization of plant extracts that are widely accepted by experts in the field, and among them, HPLC is one of the most used and accepted because its versatility and reproducibility attributes for isolation, and qualitative and quantitative estimations of active molecules (Boligon, A. A., & Athayde, M. L. (2014). Importance of HPLC in analysis of plants extracts. Austin Chromatogr, 1(3), 2).

Concerning the performing of preliminary experiments as ROS, the determination of the reactive oxygen species (ROS) will provide information about the regulating plant response to biotic and abiotic stresses, involved regulators in plant growth and development, plant hormone responses to stress (Bogatek, R., & Gniazdowska, A. (2007). ROS and phytohormones in plant-plant allelopathic interaction. Plant Signaling & Behavior, 2(4), 317-318), pro-apoptotic effects (El Khoury, M., Haykal, T., Hodroj, M. H., Najem, S. A., Sarkis, R., Taleb, R. I., & Rizk, S. (2020). Malva pseudolavatera leaf extract promotes ROS induction leading to apoptosis in acute myeloid leukemia cells in vitro. Cancers, 12(2), 435), cytotoxicity (Vallejo, M. J., Salazar, L., & Grijalva, M. (2017). Oxidative stress modulation and ROS-mediated toxicity in cancer: a review on in vitro models for plant-derived compounds. Oxidative Medicine and Cellular Longevity, 2017) and other antioxidant effects that were not considered in the objective of the present study and then, they do not result relevant for the present manuscript.

Referring to the performing of preliminary experiments as the evaluation of membrane potential, the determination of the membrane potential will provide information about the depolarizing activity of plant cells in the apoptotic process (Nadia, B., Wided, K., Kheira, B., Hassiba, R., Lamia, B., Rhouati, S., Lahouel, M. (2009). Disruption of mitochondrial membrane potential by ferulenol and restoration by propolis extract: antiapoptotic role of propolis. Acta Biologica Hungarica, 60(4), 385-398), bacterial invasion process (Ehrhardt, D. W., & Atkinson, E. M. (1992). Depolarization of alfalfa root hair membrane potential by Rhizobium meliloti Nod factors. Science, 256(5059), 998-1000) and parasitism process (Sifaoui, I., Lopez-Arencibia, A., Martín-Navarro, C. M., Ticona, J. C., Reyes-Batlle, M., Mejri, M. & Pinero, J. E. (2014). In vitro effects of triterpenic acids from olive leaf extracts on the mitochondrial membrane potential of promastigote stage of Leishmania spp. Phytomedicine, 21(12), 1689-1694) and other invasive cell effects that were not considered in the objective of the present study, and then, they do not result relevant for the present manuscript.

Reviewer 2 Report

The authors have clarified most of the issues. However, they did not answer the question, which means negative inhibition (-11%). Please explain.
In a letter to the reviewer, the authors explained the origin of the chlorogenic acid reagent. Please indicate in the text the catalog number of this reagent. You also need to correct the weight units to internationally recognized standard. Areas to be corrected are highlighted in blue.
After correcting these comments and answering the question about negative inhibition, the article can be published.

Author Response

The authors have clarified most of the issues. However, they did not answer the question, which means negative inhibition (-11%). Please explain.

Answer: Thank you very much for your comments.

First, we apologize for not including our response to your question in our previous letter to the reviewers. 

It means that at a specific concentration (8.75 µg/ml and 1.09 µg/ml) of the H. brachiata methanolic extract, no inhibition (0%) of the growth of the C. albicans strain tested was observed.  But on the contrary, the strain grew 11% more than the negative control. However, to avoid confusion, "-11%" was changed to 0%

In a letter to the reviewer, the authors explained the origin of the chlorogenic acid reagent. Please indicate in the text the catalog number of this reagent. You also need to correct the weight units to an internationally recognized standard. Areas to be corrected are highlighted in blue.

Answer: Thank you very much for your comments.

Following the instructions of the reviewer, we indicate in the text of the manuscript the catalog number of the chlorogenic acid reagent, in the Materials and Methods section.

Attending to the comments of the reviewer, we carefully corrected the weight units to an internationally recognized standard.

We hope that the additional changes to the manuscript sufficiently address the points raised by the reviewer.